# Pressure Drop Performance of Porous Composites Based on Cotton Cellulose Nanofiber and Aramid Nanofiber for Cigarette Filter Rod

**DOI:** 10.3390/ma16010411

**Published:** 2023-01-01

**Authors:** Guangyuan Yang, Ning Hou, Zheming Li, Ke Huang, Bin Zhang, Jie Xu, Jiuxiao Sun

**Affiliations:** 1China Tobacco Hubei Industrial Limited Liability Company, Wuhan 430056, China; 2School of Materials Science and Engineering, Wuhan Textile University, Wuhan 430200, China

**Keywords:** controllable porous structure, pressure drop performance, cotton cellulose nanofiber framework, ice template, aramid nanofiber

## Abstract

Porous composites have been widely used in the adsorption and catalysis field due to their special structure, abundant sites, and light weight. In this work, an environmentally friendly porous composite was successfully prepared via a facile freeze-drying method, in which cotton cellulose nanofiber (CCNF) was adopted as the main framework to construct the connected flue structure, and aramid nanofiber (ANF) was used as a reinforcer to enhance its thermal property. As-prepared porous materials retained a regulated inter-connected hole structure and controllable porosity after ice template evolution and possessed improved resistance to thermal collapse with the introduction of a small amount of aramid nanofiber, as evaluated and verified by FTIR, SEM, and TGA measurements. With the increased addition of cotton cellulose nanofiber and aramid nanofiber, the porous composites exhibited decreased porosity and increased pressure drop performance. For the CCNF/ANF-5 sample, the pressure drop was 1867 Pa with a porosity of 7.46 cm^3^/g, which best met the required pressure drop value of 1870 Pa. As-prepared porous composite with adjustable interior structure and enhanced thermal property could be a promising candidate in the tobacco field.

## 1. Introduction

Porous materials possess a huge amount of voids among solids with different shapes, dimensions, or diameters, and the micro-channels formed by the inner-connected holes all endow porous materials with special physical properties such as penetrability, pressure drop, and low density. With an appropriate choice of raw materials and preparation technology, the specific geometric parameters of porous materials can be tuned to an ideal value, finally obtaining the desired density and specific surface area, etc. [1,2,3]. Due to their high surface area, porosity, and adjustable function, porous materials are widely used in catalysis [4], separation [5], filtration [6], energy conversion [7], solar water desalination [8], and other fields. With the rapid development of material science, many new materials have been adopted to manufacture functional porous materials, while the basic application is still the transport of air or fluid.

One typical application of porous materials is the flow of smoke in the cigarette filter rod. The large-scale commercial use of cigarette filters began in 1952, using cellulose diacetate (CA), and nearly 95% of the world’s sold cigarettes are sold with a CA filter today [9]. Though CA has the advantages of a cheap price and good flue gas filtration performance, the fatal drawback of environmental pollution leads to a gloomy outlook. CA cigarette butts are the most common garbage, even in the ocean, and they are hardly biodegradable, which causes long-term harm to the environment. Several government departments and social organizations have indicated that CA cigarette butts can exist for more than ten years outside [10,11]. Thus, the recycling and reutilization of CA from discarded cigarette butts are extremely urgent. Rubén Maderuelo-Sanz et al. [12] extracted CA from cigarette butts to prepare a sound porous absorber and investigated the effect of bulk density, porosity, and flow resistivity on acoustic properties. Guo et al. [13] proposed a facile method to convert CA into a sensor for tetracycline detection and fluorescence ink with a high quantum yield of up to 22.4% and a low detection limit. Xue et al. [14] introduced recycled CA into stone mastic asphalt as a fiber stabilizer, and the modified sample showed enhanced dynamic stability and mechanical properties. However, the common recycling process is inefficient and environmentally unfriendly, and the application of recycling CA is difficult to popularize on a large scale and develop sustainably. Therefore, developing new environmentally friendly fibrous materials to replace CA may constitute a good pathway.

Cellulose, as a rising star, has been widely studied because of its wide abundance and attractive properties, in which cotton cellulose fiber accounts for approximately 40% of global fiber production [15]. Research on cellulose as a reinforcing recycled phase in nanocomposites began two decades ago, and its inherent properties, such as low toxicity, biocompatibility, and biodegradability, gave it a key role in biomaterial and other nanocomposites [16,17,18]. Mattoso et al. [19] prepared cotton cellulose nanofiber (CCNF)-reinforced thermoplastic corn starch, which exhibited enhanced mechanical properties and increased hygroscopic behavior. Liu et al. [20] synthesized carbon nanofibers/SiCN nanocomposites with CCNF as single-source precursors, and as-prepared composites possessed excellent microwave absorption performance originating from the defect-rich carbon structure. Other research on nanocellulose hybrid aerogels (SiO_2_, TiO_2_) [21,22] has reported the excellent adsorption of toxic substances from cigarette smoke. Generally speaking, rich polar groups granted it many more active sites to realize different functions or construct a stable structure and made it sensitive to humid or thermal environments simultaneously.

Aramid nanofiber (ANF), a 1D nanofiber, also has a similar structure to CA and CCNF but with a higher modulus, high-temperature resistance, and impressive chemical and thermal stabilities. Based on these advantages, it has been applied in composite reinforcement, battery separators, adsorption filtration, electrical insulation, and supercapacitor electrodes [23,24,25]. Yang et al. [26] compared CCNF and ANF in detailed experiments to eliminate the poor water resistance and poor thermal stability of CCNF via the substitution of ANF. The results declared that the ANF nanopaper demonstrated an overall improvement in performance, especially water resistance, wet strength, and thermal stability. In comparison with macroscale aramid reinforcement, ANF can easily form hydrogen bonding and π-π coordination with other substances, attributed to the ionic nature of the nanofibers [27]. In other words, ANF has a positive promoting effect on the composite interface and overall structure.

To the best of our knowledge, several new materials such as polylactic acid, cellulose nanofibers, and polypropylene have been evaluated for commercial cigarette filter rods. However, they are difficult to apply to large-scale production due to the limitation of material properties or the molding process. In this paper, a simplified binary porous system consisting of the CCNF main framework and ANF reinforcement is proposed via the freeze-drying method. By adjusting the ratio and concentration of raw materials, different porous structures and controllable pressure drop performances can be obtained to meet the performance requirements of the cigarette filter rod.

## 2. Experimental Section

### 2.1. Synthesis of Cotton Cellulose Nanofiber/Aramid Nanofiber Hydrogel

Cotton cellulose nanofiber with a diameter of 4~10 nm and a length of 1~3 μm was purchased from Guilin Qihong Technology Co., Ltd (Guilin, China). Aramid nanofiber (solid content 2.25 wt%, diameter ~40 nm) was obtained from Shandong Jufang New Material Co., Ltd (Binzhou, China). Deionized water was homemade in the lab and used throughout the whole experiment.

Cotton cellulose nanofibers with abundant polar hydroxyl groups can form a stable hydrogel with water due to a great deal of hydrogen bonding. Furthermore, the amide groups and carboxyl groups of aramid nanofiber provide convenience when combining them with cotton cellulose nanofibers. In order to guarantee good dispersion, the blending order should strictly follow the following steps. Firstly, 0.89 g of ANF was dispersed in 30 g of deionized water with a hand-held homogenizer for 5 min. Subsequently, 2.5 g of CCNF was slowly dropped into the above mixture and stirred for 15 min. The obtained hydrogel samples were sealed and stored at a low temperature.

### 2.2. Synthesis of Porous Cellulose Nanofiber/Aramid Nanofiber Cigarette Filter Rod

The above-prepared hydrogel sample was injected into a hollow round paper tube (inner diameter *Φ_inner_* = 7 mm, length *l* = 88 mm), and then transferred into a fridge at zero degrees centigrade for 12 h. Finally, paper tube samples were placed in a freezer dryer for 24 h to remove the ice template. According to the ratio and amount of CCNF and ANF, the obtained samples were labeled as CCNF/ANF-x, where the x value could be 1, 2, 3, 4, 5, or 6. The detailed composition proportion of the experimental sample and the control group sample were displayed in Table 1 with a total usage of 30 g of deionized water.

The control group samples were prepared in accordance with the process described in Section 2.1 without CCNF or ANF.

### 2.3. Characterization

As-synthesized porous CCNF/ANF composites were characterized by SEM (Sigma500, Zesis, Germany) and EDS (X-MAX, Oxford, UK) to obtain their microstructure and element distribution. FT-TR (Nicolet 6700, ThermoFisher, Waltham, MA, USA) and TGA (TGA2-LF, Mettler Toledo, Columbus, OH, USA) were used to test the chemical composition and thermal properties. The pressure drop performance of porous CCNF/ANF filter rods was recorded by the cigarette/filter rod integrated test bench (MTS-V, Chengdu Ruituo Technology Co., Ltd, Chengdu, China.) with an inlet pressure of 3 bar and a gas flow of 17.5 ± 0.1 mL/s. To eliminate the possible errors, each group tested 10 samples, and finally, exported an averaged value.

Referring to the pressure drop test standard of the CA cigarette filter rod with a standard length of 120 mm (an internal standard), the acceptable pressure drop value is 2550 ± 290 Pa. After conversion, the acceptable pressure drop value of the 88 mm long filter rod used in this test should be 1870 ± 213 Pa.

## 3. Results and Discussion

Figure 1 shows the FT-IR spectra of the pure CCNF porous sample and the CCNF/ANF porous sample. It can be observed in curve a (black line) that characteristic peaks at 3310 cm^−1^ and 1062 cm^−1^ correspond to the stretching vibration of -OH and C-O ether groups [28], and it is difficult to completely remove moisture because of the strong interaction between CCNF and water, which leads to a strong infrared absorption intensity of -OH groups. The signal located at 2908 cm^−1^ represents -CH_2_ groups, and other peaks at 1419 cm^−1^ and 1325 cm^−1^ belong to the -CH_2_ scissoring and wagging motion in cellulose. Moreover, the absorption bands at approximately 1053 cm^−1^ and 900 cm^−1^ are in accordance with cellulosic *β*-glycosidic linkages, indicating the existence of amorphous cellulose [29,30]. It is obvious that the as-prepared porous CCNF composite is merely composed of cotton cellulose nanofiber, while in curve b (red line), apart from the characteristic absorption peaks of CCNF, there are several other signals, indicating the presence of other materials. Characteristic absorption peaks at approximately 3326 cm^−1^ and 1650 cm^−1^ correspond to N-H stretching vibrations and C = O stretching vibrations from ANF [31,32]. The shift of other peaks may be attributed to the interaction of groups from two different nanofibers.

SEM images of the pure CCNF porous sample and the CCNF/ANF porous sample are displayed in Figure 2 and Figure 3, respectively. For CCNF/ANF-0 samples, a representative porous structure can be seen, and a relatively regular honeycomb appears. The section of the porous composite is approximately a rectangle, not a circle, with dimensions of approximately 200 μm. The mutually independent rectangular sections may originate from the cylindric ice template under 0 °C. After sublimation removal of the ice template, adjacent CCNF walls pull on each other and result in a tiny shape deformation. Furthermore, the disappearing ice templates retain longer internal passages, which is an advantage for gas flow and adsorption. The CCNF walls are slippery with little wrinkles. In Figure 2c,d, high-resolution section images indicate that the CCNF walls are constructed by a large number of cellulose filaments arranged in neat rows, demonstrating an excellent accumulation of hydrogen bonds. The thickness of the CCNF wall is approximately 1.5 μm, and the single CCNF diameter is below 10 nm. In addition, the EDS test of CCNF samples indicates that the weight ratio of C and O is 47.89/52.11, without other elements.

As ANF is introduced into the porous composites, tremendous changes in the micro-structure take place, as shown in Figure 3. Firstly, the smooth walls and uniform structure disappear with many tangled fibers among space or anchored on the walls, supporting more active sites for enhanced adsorption. Secondly, the pore size decreases to approximately 30 μm, and previously independent channels become interconnected, finally leading to an improved porosity. High-resolution images of rough walls show that the amount of ANF is closely associated with CCNF, and the destruction of the former regular structure can be ascribed to the stronger interaction between ANF and CCNF than the binding force between CCNF [32]. Compared with Figure 2 and Figure 3, it is obvious that ANF has been added to the CCNF main framework successfully with a uniform dispersion, which is advantageous for the mechanical property and thermal stability of the cigarette filter rod.

An EDS test of CCNF/ANF porous samples was conducted, as shown in Figure 4 and Figure 5. It can be observed that only the C element and O element exist in pure CCNF porous samples (CCNF/ANF-0), and the C/O mass ratio is approximately 47.89/52.11. Figure 5 contains three elements, namely, C, O, and N, demonstrating the existence of CCNF and ANF. These three elements uniformly spread all over the selected area, indicating excellent blending and dispersion of two raw materials in porous composites. The weight ratio of the C element, N element, and O element is approximately 72.44/8.71/18.85, and the low content of the N element indicates a small dosage of ANF. The distinct element distribution demonstrates that ANF has been introduced to the CCNF main framework smoothly.

Subsequently, the thermal stability of as-prepared porous composites was characterized by TGA under an air atmosphere. As aforementioned, aramid nanofibers have better heat resistance than cotton cellulose nanofibers, and three typical samples were chosen to make an intuitive comparison, as displayed in Figure 6. For the pure CCNF porous sample (CCNF/ANF-0, curve a), there is an initial small drop from 50 °C to 150 °C with a mass loss of 9.2%. Then the second drop begins at 210 °C and continues to 300 °C, along with the degradation reaction of CCNF chain units [33,34]. The final mass loss of the pure CCNF porous sample reaches 70%, while for the pure ANF porous sample (with 0.89 g of ANF, curve b), there is no apparent mass loss before 480 °C, showing excellent thermal stability [35,36]. Incorporating CCNF with ANF (CCNF/ANF-1, curve c), three decomposition processes can be observed with a final mass loss of 48%. The decomposition temperature of the CCNF framework increases to approximately 250 °C, and the decomposition temperature of ANF reinforcement decreases to 440 °C. As a consequence, the small addition of ANF could effectively improve the thermal resistance of porous composites.

The pressure drop performance of porous samples is listed in Table 2. Each group of porous composites was tested with 10 specimens, and all the data have been statistically processed with an outcome of an average value and standard deviation. As for CCNF/ANF-0 samples (pure CCNF porous composites), the average pressure drop is approximately 1170 Pa. With the introduction of a small amount of ANF, it increased to 1282 Pa. Upon continually increasing the amount of CCNF, the pressure drop value exhibits an approximately linear increase trend, with values of 1334 Pa, 1459 Pa, 1750 Pa, and 1867 Pa, respectively. However, in the comparison between CCNF/ANF-5 and CCNF/ANF-6 samples, an abnormal phenomenon occurs in that the pressure drop value decreases with the addition of ANF. Referencing SEM images of CCNF/ANF porous composites, this interesting phenomenon can be ascribed to the stronger interaction energy between ANF and CCNF. As the CCNF main framework is a highly complete structure in the condition of a high content, ANF can break down the existing walls that were previously separated from each other, forming an improved interconnected gas channel with a decreased pressure drop value from 2050 Pa to 1867 Pa. In contrast, the pressure drop value of CCNF/ANF-1 is higher than that of CCNF/ANF-0 samples, owing to the fact that the effective solid content is low within the porous structure. Thus, the addition of ANF could increase the pressure drop value from 1170 Pa to 1282 Pa. Furthermore, all the standard deviation values are approximately 20 Pa~32 Pa, indicating good data consistency and convincing results.

To further investigate the pressure drop performance transformation law, the inner structure and porosity should be explored in comparison with the pressure drop. The quantitative data that can be evaluated include the porosity (*P*), which can be calculated as follows [37]:(1)$$P=\frac{1}{\rho_{1}}-\frac{1}{\rho_{2}}=\frac{V}{M_{1}}-\frac{1}{\rho_{2}}$$
(2)$$V=\pi r^{2}l$$
where *ρ*_1_ and *ρ*_2_ represent the density of the CCNF/ANF porous composite and solid framework (considering the low content of ANF, we simplify it as the density of commercial CCNF, *ρ*_2_ = 1.883 g/cm^3^), *M*_1_, *V*, *r*, and *l* respectively represent the weight of the porous composite and the volume, radius, and length of the filter rod (*r =* 3.5 mm, *l =* 88 mm, *V* = 3.39 cm^3^). In addition, the weight of each single hollow round paper tube is approximately 0.47 g. Table 3 shows the porosity of as-prepared porous samples.

All the porous composites show porosity of approximately 7~13 cm^3^/g, and the minimum porosity is exhibited by the CCNF/ANF-6 sample with a value of approximately 7.34 cm^3^/g. It is apparent that the pressure drop performance shows a negative correlation with porosity in Figure 7. In other words, porous composites with a smaller porosity value display a higher pressure drop performance. A small porosity represents an unobstructed cavity structure or a low proportion of aisle space, resulting in a worse passing ability and higher pressure drop.

Therefore, the pressure drop values of CCNF/ANF-4, CCNF/ANF-5, and CCNF/ANF-6 meet the requirements of the cigarette filter rod, and CCNF/ANF-5 is the best choice. By adjusting the addition ratio of CCNF and ANF, controllable pressure drop performance with enhanced thermal stability can be obtained.

## 4. Conclusions

With a facile freeze-drying method, porous composites were synthesized and exhibited controllable pressure drop performance and enhanced thermal stability. Using ice as a soft template afforded long continuous channels for excellent gas flow, while CCNF and ANF constructed the main framework and reinforcement parts. With the increased addition of CCNF, the pressure drop value increased. ANF could break down the neighboring walls formed by CCNF, enhancing internal connectivity and providing more active sites for adsorption. With an appropriate ratio of CCNF and ANF, an ideal pressure drop could be realized. CCNF/ANF-5 showed the best pressure drop value of 1967 Pa with a porosity of 7.46 cm^3^/g. The finite element analysis of the flow field on the basis of the inner structure and porosity can be conducted in future work to deeply reveal the flow mechanism of gas in micropores. These porous nanofiber-based composites with adjustable pressure drop performances will shine brilliantly in future cigarette applications.

## Figures and Tables

**Figure 1 materials-16-00411-f001:**
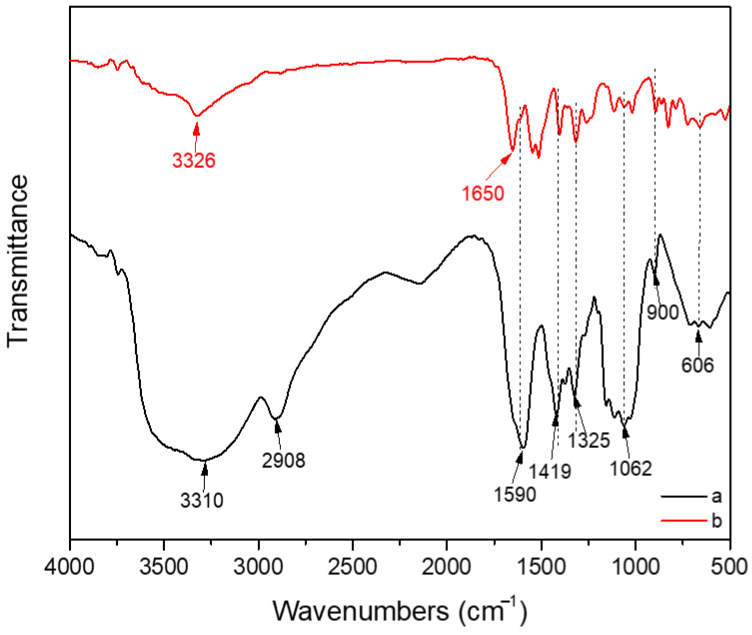
FT-IR spectra of (a) pure CCNF porous sample and (b) CCNF/ANF porous sample.

**Figure 2 materials-16-00411-f002:**
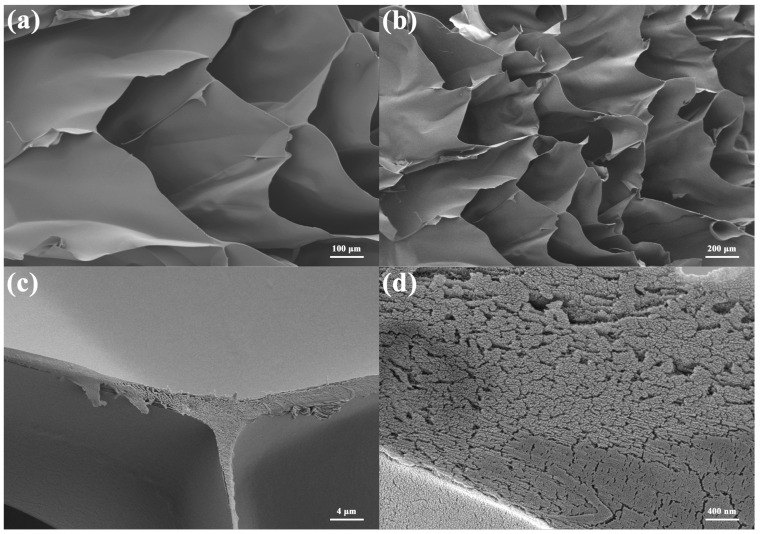
SEM image of pure CCNF/ANF-0 sample with different magnifications: (**a**) ×50, (**b**) ×100, (**c**) ×2000, and (**d**) ×20,000.

**Figure 3 materials-16-00411-f003:**
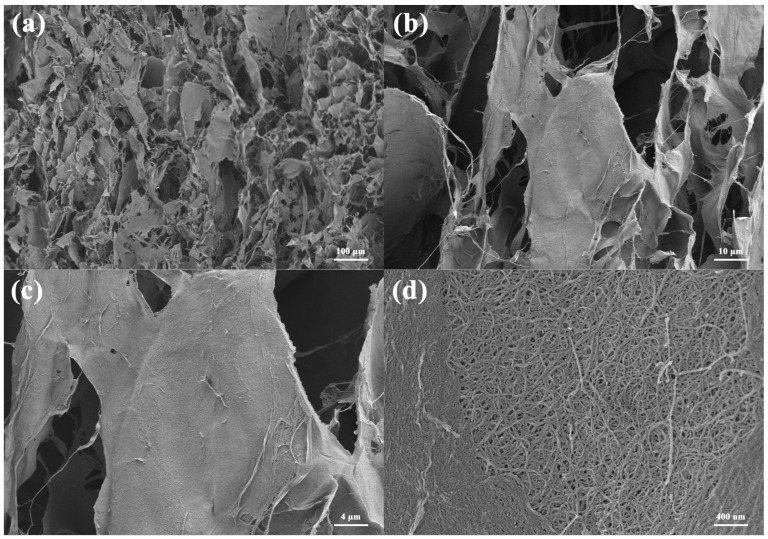
SEM image of CCNF/ANF-5 sample with different magnifications: (**a**) ×100, (**b**) ×1000, (**c**) ×2000, and (**d**) ×20,000.

**Figure 4 materials-16-00411-f004:**
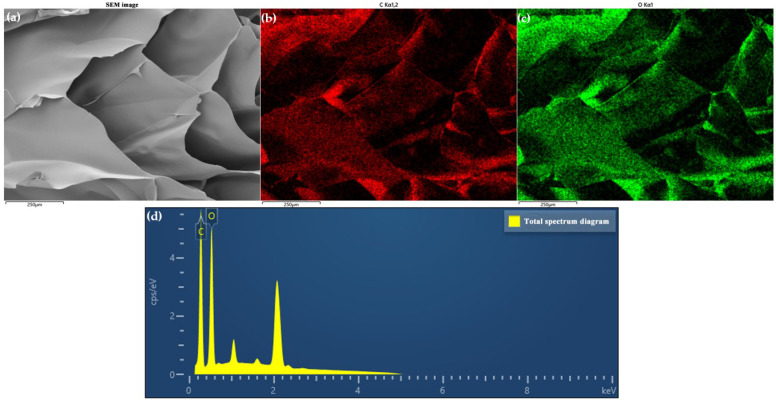
Element mappings of CCNF/ANF-0 sample: (**a**) Selective area image, (**b**) C element, (**c**) O element, and (**d**) total spectral distribution.

**Figure 5 materials-16-00411-f005:**
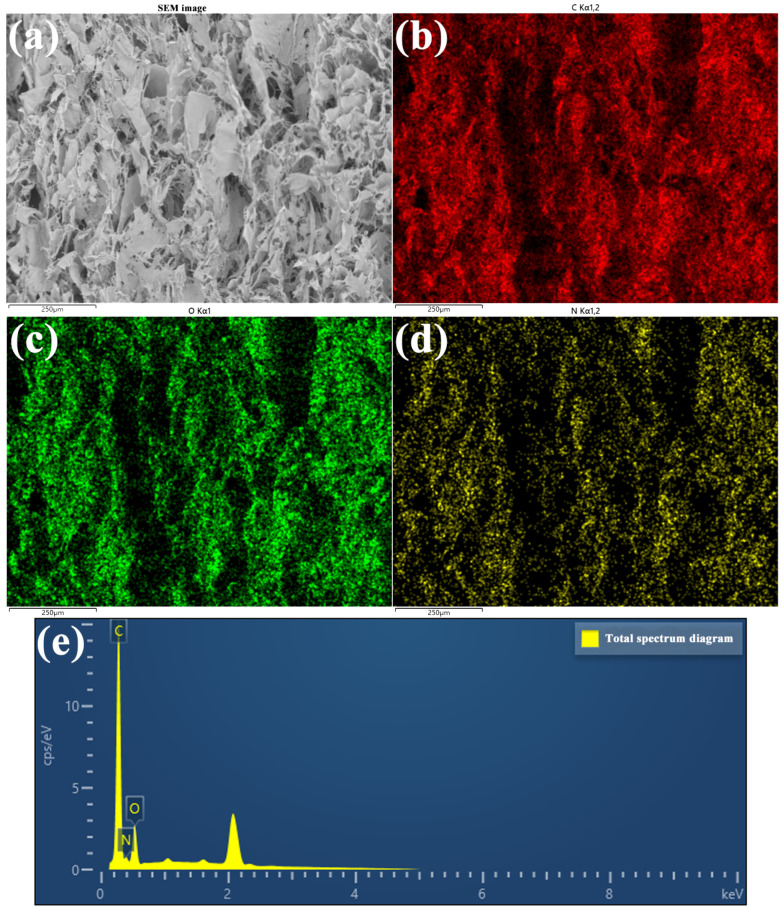
Element mappings of CCNF/ANF-5 sample: (**a**) Selective area image, (**b**) C element, (**c**) O element, (**d**) N element, and (**e**) total spectral distribution.

**Figure 6 materials-16-00411-f006:**
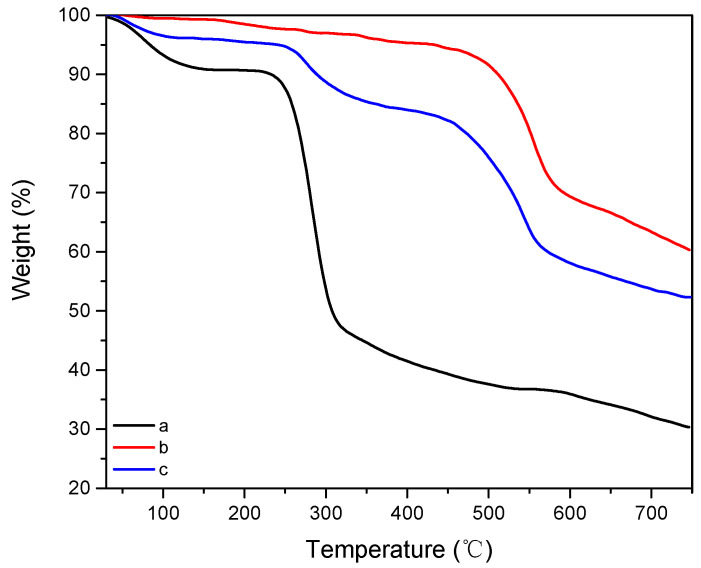
TGA spectra of (a) CCNF/ANF-0 sample, (b) pure ANF porous sample, and (c) CCNF/ANF-1 sample.

**Figure 7 materials-16-00411-f007:**
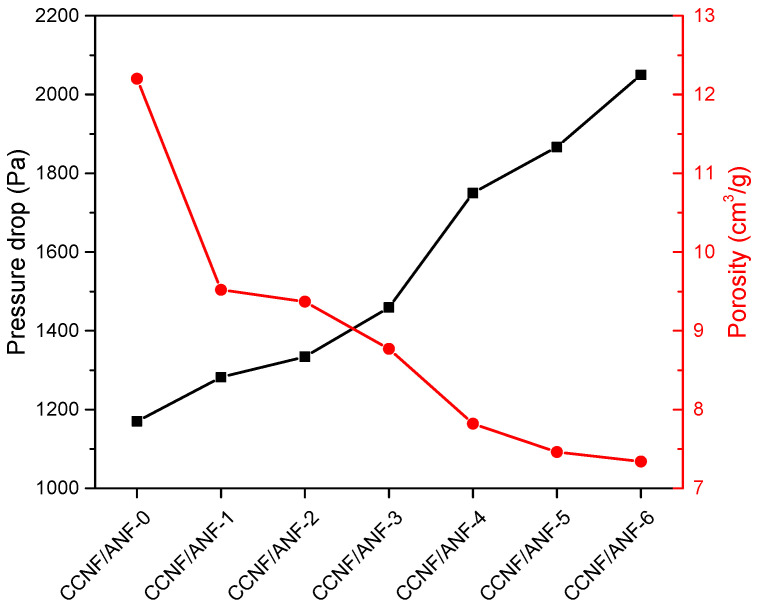
Pressure drop performance and porosity of different porous samples.

**Table 1 materials-16-00411-t001:** Composition of as-prepared samples.

Sample Name	Composition
CCNF/ANF-0	2.5 g CCNF
CCNF/ANF-1	2.5 g CCNF and 0.89 g ANF
CCNF/ANF-2	3 g CCNF and 0.89 g ANF
CCNF/ANF-3	4 g CCNF and 0.89 g ANF
CCNF/ANF-4	5 g CCNF and 0.89 g ANF
CCNF/ANF-5	6 g CCNF and 0.89 g ANF
CCNF/ANF-6	6 g CCNF

**Table 2 materials-16-00411-t002:** Pressure drop performance of porous samples.

Sample ID	Average Pressure Drop (Pa)	Standard Deviation (Pa)
CCNF/ANF-0	1170	30.50
CCNF/ANF-1	1282	32.18
CCNF/ANF-2	1334	21.37
CCNF/ANF-3	1459	24.01
CCNF/ANF-4	1750	31.81
CCNF/ANF-5	1867	25.33
CCNF/ANF-6	2050	27.84

**Table 3 materials-16-00411-t003:** Porosity of porous samples.

Sample ID	Porous Composite Weight *M*_1_ (g)	Apparent Density *ρ*_1_ (g/cm^3^)	Porosity *P* (cm^3^/g)
CCNF/ANF-0	0.266	0.079	12.2
CCNF/ANF-1	0.337	0.100	9.52
CCNF/ANF-2	0.342	0.101	9.37
CCNF/ANF-3	0.364	0.107	8.77
CCNF/ANF-4	0.406	0.12	7.82
CCNF/ANF-5	0.424	0.125	7.46
CCNF/ANF-6	0.43	0.127	7.34

## Data Availability

The data presented in this study are available upon request from the corresponding author.

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
