# Peer review of "Pressure Drop Performance of Porous Composites Based on Cotton Cellulose Nanofiber and Aramid Nanofiber for Cigarette Filter Rod"

_materials, 2023, doi:10.3390/ma16010411_

Round 1

Reviewer 1 Report

materials-2088187

Pressure drop performance of porous composites based on cotton cellulose nanofiber and aramid nanofiber for cigarette filter rod

Good meticulous work for present day research; especially as a promising candidate in tobacco field. The authors have done a worthful study in developing a porous composite made up of cotton cellulose nanofiber and aramid nanofiber.

For a curiosity, the author should answer for the below questions:

Ø  The technical properties of CCNF and ANF purchased for this study should be included in section 2.1. Either by doing the analysis or from the supplier.

Ø  Check the caption of Table 1.

Ø  What is the logic/technical reason behind selecting the ratios of CCNF and ANF.

Ø  What does 0-6 mentioned in section 2.2.

Ø  Check the labelling of SEM images, i.e., Figure 2 & Figure 3. Caption should be rewritten for both the images.

Ø  What is the major inference from Figure 2, Figure 3 & Figure 4. Author should include that in the revision.

Ø  The authors should add some recent journals to the literature review section.

Ø  References should be according to the journal template.

Ø  There are few grammatical and typing errors in the manuscript so please check and revise.

Ø  Is there any standard used for testing these properties?

Ø  There is no evident result from surface morphology results i.e., SEM and Element Mapping. It needs to be reworked.

Ø  In Figure 5, at what combination of CCNF/ANF porous samples were tested. Mention it in the figure for better understanding.

Ø  What is the major impact from facile freeze-drying method? What are the other alternatives.

Ø  How come the environmental protection factor included in the conclusion. Is there any study performed for the same?

The results and discussions are acceptable and well-presented. Include the below references to add more value for the publication; https://patents.google.com/patent/CN102423138B/en, https://doi.org/10.1080/15440478.2020.1731903, https://doi.org/10.1021/acsapm.1c01581. The technical depth is very much appropriate for the general knowledgeable individuals working for the applications of composite materials. I, as a reviewer of this manuscript, will accept this quality manuscript for publication in “Materials” after major corrections being performed from the authors.

Author Response

Thank you very much for your kind and professional suggestion, and we have modified our manuscript item by item.

Reviewer 2 Report

In this paper, Guangyuan Yang et al. report on the new strategy for fabrication of porous composite. The paper is not lacking in data. In my opinion, this manuscript needs more detailed manifestation of the novelty of the studies. In addition, I find a number of aspects are not thoroughly discussed and therefore should be included to further improve the quality of the manuscript. Please consider the following revisions.

1. Current version of the manuscript needs more detailed comparison of obtained results with the literature data, It will improve the quality of the manuscript considerably. For example, below is several articles that can be included to improve introduction.

10.1021/acs.nanolett.1c04216

https://doi.org/10.1016/j.polymer.2018.04.064

10.1021/acs.nanolett.1c02593

2. What is the accuracy of porosity value (Fig. 6)?

3. Please indicate opportunities to increase scalability of composite fabrication.

4. I have small remark (that can be improved at the proofreading, if it is needed). It is worthwhile mentioning in text, that a vapour/solid interface (that is very important for porous material) have confined water which affect on the properties of material, such as thermodynamic or transport (see 10.1021/acs.chemrev.6b00045 , https://doi.org/10.1007/s10765-013-1552-6 ). It is important for understending and development of mesoporous materials.

I should like to recommend publication of this article in MDPI magazine after minor revision.

Author Response

(The authors gave the same response as above.)

Reviewer 3 Report

 Accept after minor revision (corrections to minor methodological errors and text editing)

Author Response

(The authors gave the same response as above.)

Round 2

Reviewer 1 Report

materials-2088187

Pressure drop performance of porous composites based on cotton cellulose nanofiber and aramid nanofiber for cigarette filter rod

I have gone through the manuscript completely and satisfied with the revision. As such, I will recommend this paper for publication in its present form.